# Between Privilege and Oppression: An Intersectional Analysis of Active Transportation Experiences Among Washington D.C. Area Youth

**DOI:** 10.3390/ijerph16081313

**Published:** 2019-04-12

**Authors:** Jennifer D. Roberts, Sandra Mandic, Craig S. Fryer, Micah L. Brachman, Rashawn Ray

**Affiliations:** 1Department of Kinesiology, School of Public Health, University of Maryland, College Park, MD 20742, USA; 2Active Living Laboratory, School of Physical Education Sport and Exercise Sciences, University of Otago, Dunedin 9054, New Zealand; sandra.mandic@otago.ac.nz; 3Department of Behavioral and Community Health, School of Public Health, University of Maryland, College Park, MD 20742, USA; csfryer@umd.edu; 4Center for Geospatial Information Science, Department of Geographical Sciences, College of Behavioral and Social Sciences, University of Maryland, College Park, MD 20742, USA; brachman@umd.edu; 5Department of Sociology, College of Behavioral and Social Sciences, University of Maryland, College Park, MD 20742, USA; rjray@umd.edu

**Keywords:** active transportation, PEAT Study, transportation inequity, youth of color

## Abstract

The use of active transportation (AT), such as walking, cycling, or even public transit, as a means of transport offers an opportunity to increase youth physical activity and improve health. Despite the well-known benefits of AT, there are environmental and social variables that converge on the AT experiences of low-income youth and youth of color (YOC) that have yet to be fully uncovered. This study uses an intersectional framework, largely focusing on the race-gender-class trinity, to examine youth AT within a context of transportation inequity. Theoretically guided by the Ecological Model of Active Transportation, focus groups were completed with two groups of girls (15 participants) and two groups of boys (nine participants) ranging between the ages of 12–15 years who lived within the Washington D.C. area. This research found race, gender, and class to be inhibitors of AT for both boys and girls, but with more pronounced negative influences on girls.

## 1. Introduction

### 1.1. Transportation Inequity

*“Nationally, the United States remains a country where many forms of transportation are effectively still segregated—whites and minorities ride different kinds of transportation, resulting in an unequal ability to reach jobs, education, and a better life”*. Corinne Ramey, “America’s Unfair Rules of the Road” [1].

Transportation is an essential element in our daily lives. While transportation justice aims “to equally and equitably address the [transit] needs of all people, regardless of economic class, race, sex, age, ability or any other kind of social distinguisher,” the United States endures considerable inequities in how individuals traverse [2]. In the United States, children and adolescents from low-income or historically disadvantaged minority families experience more transportation obstacles, which not only profoundly restricts their access to basic needs, but also creates a perpetual impoverished entrapment that limits their upward mobility [3,4,5,6,7]. Youth from low-income families and youth of color (YOC), particularly African Americans and Latinxs, are more likely to bicycle or walk to school compared to their White counterparts or higher income youth [5]. A high percentage of public transportation users are from low (<USD $15,000) to moderate (USD $15,000–$50,000) income families and African Americans make up 33% of the ridership [8,9]. Since parents and caregivers of low-income youth and YOC generally have more limited resources to own, fuel, maintain, or access a personal vehicle, walking, bicycling and public transportation are the primary modes of transport that are used by these youth [6,10]. While income and race appear to be independent determinants of transport options, this research will begin to uncover the complex intersectional influences of these and other modalities of identity on transit choice for low-income youth and YOC.

### 1.2. Overweight/Obesity and Physical Inactivity

In the United States, African American, Latinx and low-socioeconomic status youth have been found to be disproportionally affected by the childhood obesity epidemic [11,12]. Specifically, in 2017 the obesity prevalence rates among African American (22.0%) and Latinx (25.8%) youth aged 2–19 years were higher than among White (14.1%) youth [12]. Additionally, obesity prevalence rates were highest among youth in families with an income-to-poverty ratio of 51–100% or less [12]. A deeper intersectionality of race, ethnicity, gender and income with obesity has also revealed interesting findings among these children and adolescents. For example, Mexican American boys (40.5%), compared to White boys (34.5%) or African American boys (32.1%), and African American girls (44.5%), compared to White girls (31.7%) or Mexican American girls (37.1%), had the highest obesity prevalence rates based on National Health and Nutrition Examination Survey data [13]. Furthermore, socioeconomic status has been found to be inversely related to obesity in White youth, but not in African American or Mexican American youth [13]. An inverse relationship with socioeconomic status has nevertheless been recently identified in other youth Latinx groups [14]. Finally, the 2018 United States Report Card on Physical Activity for Youth and Children reported that African American and Latinx youth were more sedentary and much less likely to meet the screen time guideline of two hours or less per day compared to White youth [15,16]. While the majority (76%) of American youth 6–17 years old do not meet the minimum Physical Activity Guidelines (at least 60 min of moderate- and vigorous-intensity physical activity per day), variability in physical activity levels by intersections of race, ethnicity and gender have been identified [16]. Unlike White girls, African American and Latina girls are significantly less physically active compared to their boy counterparts [17] with African American girls reporting 27 min of moderate-to-vigorous physical activity per day less than their boy counterparts [17].

### 1.3. Active Transportation

Active transportation (AT), such as walking, cycling or using public transit, as a means of transport offers an opportunity to move closer to reaching the Healthy People 2020 objectives of “reducing the proportion of children and adolescents who are considered obese” and increasing physical activity [18,19]. Youth that use AT have better cardiorespiratory and muscular fitness, increased energy expenditure, more favorable body composition and less weight gain compared to youth that do not engage in AT [20,21]. While AT can be used as a strategy to address the physical inactivity and overweight/obesity public health issues affecting low-income youth and YOC, a fundamental paradox exists between their use of AT and the potential negative impacts of AT engagement on their health and well-being.

In addition to individual factors, environmental, social and policy factors influence youth AT patterns [22,23,24,25,26]. Factors such as neighborhood street connectivity, land use mix, urbanization, family time constraints, youth fear coping, and parental risk perceptions can positively or negatively influence youth AT [27,28,29,30]. For example, a lack of pedestrian and/or cycling infrastructure, concerns for personal safety, micro-aggressions or harassment from police officers, street trash and debris, or other environmental and social barriers, will make youth less inclined to engage in purposeful or intentional AT [27,28,29,30]. Youth from low-income families and YOC are often relegated to travel on foot, by bicycle or using public transport and do not necessarily “choose” AT as their main transport mode. As such, this inequity in transportation choice often discourages purposeful AT or engagement of AT that is a preferred decision. While other demographic groups may myopically view the benefits of AT, there are environmental and social variables that converge on the AT experiences of low-income youth and YOC, thus making AT a positive, negative, or in some cases, fatal (e.g., “Walking While Black”) experience [31,32]. To examine the AT experiences of low-income youth and YOC, the demographic, environmental and social intersectional influences on youth AT need to be fully explored and considered in order to design fully inclusive interventions for AT promotion.

### 1.4. Intersectionality Framework

Intersectional analysis is often viewed as a framework for deconstructing power relations within society, however, the derivation of the framework arose in opposition to theoretical mainstream conceptualizations of identity [33,34]. Following the struggles of the Civil Rights and Women’s Rights Movements, feminist scholars argued that race, gender, and class were interconnected as “intersecting oppressions” [35] and that race and gender “are not separate and additive, but interactive and multiplicative in their effects” [36]. Figure 1 illustrates the intersecting axes of privilege, domination and oppression [37].

The application of intersectional theory to AT research has the potential to contribute significantly to the overall public health knowledge base. These theoretical offerings include recognizing inequalities with precision, developing intervention strategies, and ensuring the identification of community-specific and community-relevant findings [38]. Although recently noted as an important theoretical framework for public health analyses, intersectionality has been explicitly incorporated into feminist academic work for over twenty years [39]. As public health scholars call for greater theorization of analyses, there is a profound level of unrealized and untapped potential to construct theoretical and methodological bridges between intersectionality and AT research. For example, when physical environment, neighborhood or place based research is examined, it is recommended that the intersectionality paradigm place the lens on power, oppression and difference [40]. Specifically, a mindfulness of the multiple and co-existing systems of oppression influenced by identities is a fundamental and essential component of any research. As such, this involves an examination of race, gender, class, and subjugative modalities, which position individuals and social groups in shifting positions of privilege and/or disempowerment [41].

Race-gender-class has been the traditional trinity of intersectional studies, however, intersectionality applications vary from unitary to multiple approaches [38,42]. When only one modality or social category of position, such as race, is examined, a unitary approach is practiced [42]. With a multiple approach, more than one social category is of interest, which then operates under an additive assumption whereby each modality of marginalization or privilege is layered to summation [42]. An examination of our research has taken a broader approach and considers the traditional race-gender-class trinity of identity within a context of transportation inequity. Similar to Diez-Roux et al. (1997), who applied intersectionality theory after observing interactions between race, gender and class within their neighborhood-based study of coronary heart disease, it is theorized that there will be some effects on AT behaviors and attitudes in some population groups and in some type of manner [43]. Administering an intersectional framework to AT research has important implications for the literature. As AT involves the human movement from one place to another or from one neighborhood to another, the power dynamics that flow from race, gender, class, and other systems of subjugation or privilege will generally transcend the boundaries of any given space, place or neighborhood [44]. Therefore, the AT field is encouraged to move beyond the status quo and embrace the concept of “relational geographies” in order to stimulate “a more sophisticated perspective” involving “more analysis of how human and physical phenomena need to be understood as an outcome of interrelated processes [operating] simultaneously at various spatial scales” [40]. Furthermore, the application of an intersectional paradigm to AT research further supports exploratory work examining how historical and contemporaneous power dynamics impact communities and individuals, particularly youth, in spatial settings. Therefore, the purpose of our analysis, was to apply an intersectionality framework using the race-gender-class trinity in order to explore the AT experiences of a population of Washington D.C. area youth within the context of transportation inequity. Specifically, transportation inequity was defined as any barrier that inhibited movement or prohibited access into or through spaces due to multiple power dynamics intersecting with individual, sociocultural, environmental and policy influences of youth AT. This approach offers a significant and essential change to the conformity with which the AT and public health literature is currently positioned.

## 2. Methods

### 2.1. Physical Environment and Active Transportation Study

With a new opportunity for AT in Fairfax County, Virginia, the Physical Environment and Active Transportation (PEAT) Study explored the impact of the Silver Line-1 on youth AT while also examining the demographic, environmental and social intersectional influences on youth AT. The Silver Line-1 (Dulles Corridor Metrorail Project (Silver Line-First Phase)) of the Washington, D.C. Metropolitan Area Transit Authority (Metro) system opened on 26 July 2014 with a total of 28 stations, including five new stations in Fairfax County (Figure 2). In an area lacking heavy rail transportation options within the urban fabric of the region, this Metro system extension provided a new physical environment and an opportunity for Fairfax County youth to engage in AT to and from Metro stations on their way to many destinations. In Fairfax County, the childhood overweight/obesity prevalence rate is approximately 26% and only 21% of the youth have achieved the daily minimum physical activity recommendation [45,46]. While there are disparate rates of youth overweight/obesity by race and ethnicity within the entire state of Virginia (White: 24.3% vs. African American: 33.9% vs. Asian: 17.0% vs. Latinx: 35.9%), disparity rates of physical inactivity among adolescents have also been found by gender (males: 67.7% vs. females: 82.4%); race and ethnicity (White: 70.6%; African American: 79.0%; Asian: 84.0%; Latinx: 82.7%) and by increasing age [47]. To this end, the PEAT Study was guided by two research objectives. First, with the introduction of the Silver Line-1, the implications of this new physical environment on Fairfax County youth AT were explored. Second, through this exploration in an area representing inequity of transportation choice, an intersectional approach, largely focusing on the race-gender-class trinity, was applied in order to understand the inequalities that influence youth AT engagement.

### 2.2. PEAT Study Design

The Ecological Model of Active Transportation theoretically guided the design of the PEAT Study (Figure 3) [48]. Specifically, this multi-level model moves beyond the individual (e.g., convenience) domain and considers other domains of influence on AT choice, such as the built (e.g., presence of sidewalks), social (e.g., parental rules) and policy (e.g., transportation school policy) environments. To gain an in-depth understanding of the lived experiences and underpinning decisions influencing youth AT behaviors, attitudes and perceptions, the PEAT Study employed a qualitative design using semi-structured and gender-stratified focus group discussions (FGD) with adolescents aged 12–17 years. FGD was chosen as the preferred qualitative method because it: (1) provides an opportunity to gather a breadth of data about the phenomena under study; (2) has the ability to simulate group and real-world dynamics; and (3) operates on peer-to-peer interactions [49]. 

### 2.3. Study Site and Population Recruitment

Participants were adolescents who lived within a two-mile buffer of the five new Silver Line-1 Metro stations. Partnership with Reston Community Center facilitated study participant recruitment. Reston Community Center permitted usage of their space and recruitment of their members, as well as, members of the Reston Teen Center and Southgate Community Center, which included any and all Reston residents. All three recruitment sites were located in Fairfax County and within a two-mile buffer of the Wiehle-Reston East Silver Line-1 Metro station, the station located within Reston (Figure 4). 

### 2.4. Focus Group Discussions

FGDs were conducted between May and August 2016. Purposeful sampling consisting of gender-specific FGD (4–10 adolescents/group) was implemented until saturation (no additional information was observed) [50]. Each FGD session lasted approximately one hour and was moderated by one of two trained researchers. Participants received $10 dollars for participation.

Prior to the commencement of each FGD, youth participants and their parents were informed of the PEAT Study and provided with information sheets and brochures. Through an opt-out mechanism, informed consent was obtained from the parents. Hence, we obtained parental consent for all adolescents participating in the FDGs. At the start of the discussions, the moderator explained the study purpose and procedures. All youth participants then completed their assent forms. Pseudonyms were chosen by each participant to not only maintain anonymity, but to also begin each FGD with a bit of creative fun.

A semi-structured FGD guide addressing all domains of the Ecological Model of Active Transportation was used to facilitate the discussions. This guide posed open-ended questions to foster a free-flowing and flexible conversation regarding AT behaviors and attitudes. Probes and clarifying questions were used as needed to expound on the experiences and ideas of youth. FGD data were digitally recorded, transcribed verbatim by two transcriptionists, manually coded, and analyzed. 

### 2.5. Qualitative Data Analysis

Data were analyzed using a content analysis technique, which identified emergent themes or trends from transcripts [51]. The coding procedures followed a series of steps which first included the development of a set of preliminary codes (themes) corresponding to the individual, sociocultural, environmental and policy domains of AT in the FGD guide. Specifically, all FGD transcripts were read and reviewed by pairs of PEAT Study researchers who identified and generated initial codes, concepts and themes independently. Subsequently, a meeting was held with researchers and the Principal Investigator (JDR) to review, define and name the prominent themes [52]. Additional codes were created that arose during the meeting and that were of special interest. Next, non-substantive codes (e.g., illustrations of particular points) were developed which were used to produce subsequent detailed codes for analysis of specific topics. For these final steps, computer-assisted qualitative data analysis software, NVivo 11, was used to assist with the coding process. This inductive analysis considered the individual contributions of participants as well as the group interactions. Finally, an intersectionality framework examining the demographic, environmental and social intersectional influences on youth AT was incorporated within the analysis. Content identified themes related to AT behaviors and perceptions were analyzed through an intersectional race-gender-class trinity while also examining other modalities of identity relating to transportation inequity.

### 2.6. Research Ethics

The Institutional Review Board at The University of Maryland at College Park (UMD) approved this study protocol (UMCP, 819986-3). Data analysis and interpretation occurred at UMD School of Public Health.

## 3. Results

FGDs with two groups of adolescent girls (*n* = 15) and two groups of adolescent boys (*n* = 9) were conducted on three different dates between May and August of 2016 at the Reston Teen Center. On the basis of cross-race effect, phenotypic identification, as well as the provided community center’s membership profiles, over 90% of participants were categorized as YOC, specifically African American and Latinx [53,54]. The participants age range was 12–15 years with an average of 13 years.

Themes related to youth AT behaviors and perceptions were identified within the analytic framework. Verbatim quotes (VQ) illustrating emergent themes have been labeled by the FGD gender and number (e.g., GG1 = Girl Group 1; GG2 = Girl Group 2; BG1 = Boy Group 1; BG2 = Boy Group 2). Additionally, participant responses (PR) to other participant quotes were labeled accordingly whereby each letter (e.g., PR_A) represents a response from a different participant. For this analysis the pseudonyms have been omitted.

### 3.1. Active Transportation Behavior Themes

Many of the participants used AT to travel short-to-medium length distances (e.g., 10–20 min). The most common forms of AT were walking and public bus use. Emergent themes related to AT behavior included (1) family influence and (2) friend influence, as well as (3) incentives, and (4) disincentives of AT behaviors. Similar to prior research, participants’ AT behaviors were found to be highly influenced by family and friends among both boys and girls in this study [27,28,29,30].

VQ 1*“my friends want to walk everywhere and I don’t know why”* (GG1)PR_A: *“whenever you come over to my house we always walk and I ask you and you say lets walk, and that’s influencing”**“me?”* (GG1)   PR_A: *“yeah you”*VQ 2*“I use the school bus when I’m at my Dad’s house, so Mondays, Tuesdays and Fridays”* (GG2)VQ 3*“Oh, my mom drives me around a lot. My brother, he’s twenty-eight, he can drive me”* (BG1)VQ 4*“Well, it’s what you just said. So if it’s a nice day, I just say I’ll walk instead. Or at least my mom forces me to walk”* (BG2)

In performing a query matrix by text frequency, the most prevalent sub-themes of AT behavior incentives and disincentives were related to physical activity, passive transportation, safety, parent rules and convenience. Expressly, participants reported exercise as an incentive to engaging in AT. 

VQ 5 *“…since I’m trying to work out more I sometimes walk over to the Safeway or something or walk over to like a certain place that she* [Mom] *needs to go”* (GG2)   PR_A: *“I walk my dog for one hour”*    PR_B: *“I like take the trails and stuff and do walking in different parts and then um sometimes I* [*run*] *but that’s rare because running is hard.*VQ 6*“If you walk, like maybe a mile or two to the nearest grocery store, you lose calories”* (BG1)   PR_A: Participant provided a “fist bump” hand gesture 

Passive transportation, or any transportation solely by personal vehicle was found to be the most frequently discussed disincentive to AT, followed by safety, parent rules and convenience. Strong habitual behavior, such as driving and ingrained social norms can favor passive transportation over various modes of AT [55,56].

VQ 7*“Driving is our main priority, but if something happens to that car, we take the bus to the metro”* (BG1)

Safety was discussed in terms of victimization as well as personal injury due to a lack of sidewalks or pedestrian crossings. Moreover, safety was often discussed through the lens of parent rules or safety concerns for risk of stranger danger and injury. 

VQ 8 *“My Mom doesn’t let me go anywhere alone”* (GG2)PR_A: *“Um, my Mom always tell me to bring your phone and she usually doesn’t want me to go anywhere else that’s far away. She usually, well my brother come along with me. Like when I* [go to the] *snack machine down there from my house she did not like it, she always wants me to ask my brother to come with me.”*    PR_B: *“Um, my Mom’s kind of both. She likes me walking places for exercise but she doesn’t like um being afraid that I could get in trouble or I could get kidnapped or something so she usually drives me around places”*VQ 9 *“Um, I like sometimes like start walking somewhere or I think about walking somewhere and get ready for it but then I start thinking about which path I’m gonna take and which people are there cause sometimes there are those little tunnels that are under the street and stuff and there’s suspicious people there so my Mom’s like oh, I’m not sure if you should go that way so then I either change to do it later like I do something else”* (GG2)PR_A: *“Um I want to walk to Shoppers, but my Mom say no because I’m too small to go to Shoppers. Usually my brother goes.”*VQ 10*“*[Sidewalks] *Always clean? No”* (GG1)PR_A: *“Sometimes they have sidewalks on one part of the road but not the other so you would have to bike up the dirt and grass or something.”**“Oh sometimes when your crossing the street they don’t have lights on both sides, you know they could be going this way but there could only be a light going that way.”* (GG1)

Convenience was mostly related to destination distance, weather and time constraints. Furthermore, participants expressed the convenience of AT behaviors as a function of personal motivation.

VQ 11*“…or run or bike. Anyway I can…if there’s a close bike ride there”* (BG2)VQ 12 *“Rain, that’s one thing. Because who wants to walk”* (BG1)PR_A: *“Yeah, when it’s raining, I can just wait inside the house until the bus gets here, just walk outside.”*VQ 13*“…um, I’ve started walking to the pool and then I had to go back because it started raining and before my stepdad has told me that it’s too hot to walk anywhere and you’ll have a heatstroke”* (GG2)VQ 14*“Yeah it’s like a 30 min walk”* (GG1)PR_A: *“…period cramps Man you don’t want to go through any of that and you just want to eat and sleep and you don’t want to do anything.”*VQ 15*“…like after school there’s this like shopping center that’s close so then* [friends] *like encourage me to walk there but since I really am like lazy sometimes I don’t feel like walking”* (GG2)

### 3.2. Active Transportation Perception Themes

Coding references (CR) within the data were classified into positive, neutral and negative AT perception themes. Using the number of CRs across all participants, positive (CR = 10) AT perception was the most prominent, followed by negative (CR = 7) and neutral (CR = 4) AT perception themes. Sub-themes of positive AT perception were related to enjoyment, environmental benefits, and physical activity while negative AT perception sub-themes were affiliated with issues of inadequate built environment features, stranger danger and discomfort.

VQ 16 *“I do* [AT] *all the time. It’s all fun to do.”* (BG1)PR_A: *“If you walk, like maybe a mile or two to the nearest grocery store, you lose calories”*VQ 17 *“…trying to be more active for the environment…and help with environment and pollution and stuff like that and health-wise”* (GG1)VQ 18*“…because like there’s no sidewalks on one of the streets.”* (GG2)VQ 19*“…all of these people being kidnapped and raped and murdered and stuff like that *[Mom]* automatically assumes that even if I walk for 5 min”* (GG1)PR_A: *“my parents are divorced so my mom isn’t okay with* [AT] *but my dad is”*VQ 20*“Like biking for fun is fine but I don’t like trying to bike somewhere because all those hills and stuff I just don’t like it”* (GG1)PR_A: *“So someone could steal your bike at the Metro because you know it can get crowded and like even if you lock it up you got those crooks that unlock your bikes”*

The type of AT (e.g., public transit vs. walking) was echoed as a neutral AT perception sub-theme, which indicated that certain forms of AT were more favorable in comparison to other forms. Additionally, engaging in AT was conditional of companionship for some participants.

VQ 21*“a bus goes faster than like feet, cause I take 40 min to walk to school…”* (GG2)PR_A: *“Um, the bus I take is a longer ride because I live in Centreville, and my school is in Clifton and its very far away, but my Mom told me that my brother and my sister, they once ride a bike to school and I don’t want to try”*VQ 22*“as long as I’m with one other person then* [AT] *is okay”* (GG1)   PR_A: *“I have to be with 15 people”*       PR_EVERYONE: Laughter

### 3.3. Intersectionality of Active Transportation Themes

When AT behaviors and perceptions were intersected with gender, race and class, varying expressions of AT were observed. Categorically, gender influenced AT behaviors and perceptions either directly or through the meniscus of parents, friends and/or societal expectations. In this study, boys were more likely to bike for transport and express enjoyment or pleasure with this form of AT compared to girls (see VQ 16; VQ 24; VQ 25). Furthermore, adolescent boys were often encouraged (see VQ 4) by parents, relatives, and friends to engage in AT, which was not conditional to companionship (see VQ 25). In contrast, adolescent girls were discouraged from engaging in AT and any permissible AT was in fact conditional to companionship (see VQ 8; VQ 22). Therefore, adolescent girls unequivocally reported a fear of crime and specifically stranger danger (see VQ 8–9; VQ 19; VQ 23;).

VQ 23*“My mom always tells me it’s not safe, I can’t be walking alone”* (GG2)PR_A: *“Um sometimes I don’t feel safe walking like by myself or with other people cause of like cat-callers”*VQ 24*“I bike a lot, go out in neighborhood. I walk a lot too… Just around the neighborhood. It’s fun”* (BG1)PR_A: *“Walking, it’s what I mostly do. I used to bike, but…yeah”*    PR_B: *“Yes,* [I do AT because it’s fun] *and I like walking. My mom doesn’t like walking for some reason”*VQ 25*“I hop on my dirt bike and drop myself* [off]*”* (BG2)

Importantly, when the aforementioned CRs were stratified by gender, these oppositional observations among girls and boys were more apparent. For girls, negative AT perception (CR = 7) was the most prominent theme, followed by positive (CR = 6) and neutral (CR = 4) AT perception themes. Conversely, the boys only expressed positive (CR = 4) AT perceptions and no neutral (CR = 0) or negative (CR = 0) AT perceptions.

Perceptions of racism were reported throughout the course of these FGDs. Over half of the participants described not feeling safe or welcomed because of the color of their skin. This consciousness, however, was only explicitly reported by the adolescent girls in this study. Subtle comments regarding the safety of walking in “certain areas” or around “certain people” were offered by the adolescent boys whereby such comments can be open to interpretation.

VQ 26*“well people* [are] *not safe to be honest they’re really mean…the neighborhood I live in they’re really racist because one time one of my friends were riding my bike and she actually went on his yard and he yelled at her for like no reason and he’s like get the blank off my yard like what the blank are you doing, stuff like that…Yeah,* [he said something racist]*”* (GG1)      PR_EVERYONE: Nervous LaughterVQ 27*“Yeah, it’s better to not walk at night”* (BG1)PR_A: *“*[It is not safe to walk] *“uhhh…certain, like…areas certain people live”*

In previous studies, neighborhood socioeconomic status and the perception of safety have been inextricably linked with physical activity [57,58,59,60]. Specifically, perceived neighborhood safety concerns, particularly among women, girls, and those living in low-income communities represent a barrier to physical activity and thus AT [57,58,59,60]. The intersectionality of socioeconomic class and AT revealed through the qualitative findings of this study further confirmed these established observations. Specifically, AT behaviors were highly influenced by socioeconomic status. Thus, the engagement of AT may have occurred due to the availability or lack of certain options and resources for alternative modes of transport. 

VQ 28*“I mean my neighborhood’s pretty safe so really like everyone else in our neighborhood is like up here on the money scale and we’re like down here so”* (GG1)PR_A: *“I have to cross a bunch of streets and it’s like in a suspicious neighborhood”*VQ 29*“Yeah. Or like when I sometimes like try* [to AT] *and* [they] *want me to go bike riding with them but I don’t have a bike, but they keep forgetting that.”* (GG2)VQ 30*“I’d have to take the bus, like the public bus to school because my Mom didn’t have a car”* (GG2)

## 4. Discussion

On the evening of February 26, 2012 in Sanford, Florida, a 17-year old boy was walking back to the home of his father’s girlfriend after a trip to the nearby convenience store carrying a small bag of candy and a can of juice while speaking to his girlfriend through a hands-free earpiece on his cellphone. A self-appointed neighborhood watch volunteer who was patrolling the community in his vehicle spotted this boy and called the Sanford police to report a “suspicious person” in the neighborhood. Despite being instructed by the dispatcher to not leave his vehicle or follow the teenager, he did so and fatally shot this boy with a handgun [31,32]. This boy was using AT. He was African American. His name was Trayvon Martin.

### 4.1. Active Transportation—An Unwelcomed Choice

Although many individuals solely highlight the positive advantages of AT, such as the health (e.g., increased physical activity) and environmental (e.g., reduced vehicular pollutants) benefits, there are other variables that can negatively, and even lethally, impact the AT experiences of low-income youth and YOC. Unfortunately, the case of Trayvon Martin is not unique. The expression “Walking While Black”, a double entendre derived from the United States criminal offense of “driving while intoxicated”, is a manifestation of the racial profiling endured by many African American pedestrians. It is thought that the “Walking While Black” expression originated from a 1999 New York Times article stating that “his [Mayor Rudolph W. Giuliani] numbers, however accurate, cannot reflect the anger felt by people stopped and frisked by the police each day only because they are W.W.B.—Walking While Black” [61]. At that time, African American New York City residents were infuriated by Mayor Rudolph W. Giuliani and Police Commissioner Howard Safir for questionable shootings and being racially profiled by the New York Police Department. Even though the expression has expanded to include other similar forms of aggression and harassment (e.g., “Driving While Black”; “Traveling While Black”; “Shopping While Black”), the original “Walking While Black” has most recently recognized another nuance to the expression. Pointedly, in the United States, people of color are disproportionately affected by pedestrian injuries and deaths. Throughout the course of a decade from 2001 to 2010, African American and Latinx men were more than twice as likely than White men to die from pedestrian crashes and American Indian/Alaska Native men had a rate more than four times higher [62]. Two recently published studies conducted in Portland, Oregon and Las Vegas, Nevada have examined the influence of racial bias on driver yielding behaviors as a potential contributing factor to the higher rates of pedestrian crashes affecting people of color [63,64]. In Portland, African American pedestrians were passed by motorists twice as many times and experienced wait periods that were 32% longer than White pedestrians [63]. In Las Vegas, more cars passed through a crosswalk while an African American pedestrian was in the roadway compared to a White pedestrian in a high-income area of the city [64]. These data demonstrated that walking as a form of AT for African Americans may put them at an additional risk to injury or death simply due to the color of their skin. While both of these studies were conducted on adults, our research findings, informed by an analytic paradigm attuned to the intersection of race, offers a perspective to the risk of microaggressions and pedestrian fatalities when engaged in AT directly from the voices of African American and Latinx youth. Interestingly, the adolescent girls in our study most clearly and explicitly articulated this perspective even though the number of boys and men currently experiencing these types of hostilities in the United States is overwhelming. This may speak to the notion that African American and Latinx boys are misunderstood as threats to seize and neutralize whereas African American and Latinx girls are perceived as nuisances to besiege and verbally assault.

### 4.2. Boys of Color vs. Girls of Color

PEAT Study participants were residents of Reston, a planned community located in Fairfax County, Virginia. With a population slightly above 60,000 according to the most recent United States Census, the composition of African American and Latinx residents is 22%. Although Reston is situated within the largest metropolitan region (Washington-Arlington-Alexandria area) of the entire State of Virginia, the continued residential segregation is still very prevalent. In 2012, the dissimilarity index, the most common way of measuring residential segregation between African Americans and Whites throughout neighborhoods, for the Washington-Arlington-Alexandria area was reported to be 62 out of 100 [65]. A dissimilarity index of 0 indicates that African Americans and Whites are evenly distributed across census tracts in comparison to an index of 100, which indicates that African Americans and Whites live in completely separate neighborhoods. With an index of 62 in this Northern Virginia area, it can be adjudged that most, if not all, of the PEAT Study participants live in predominately African American and Latinx neighborhoods.

The perception of African American and Latinx neighborhoods for its residents can be one of a village for boys or a trouble spot for girls. For example, the men in the neighborhood may be viewed as uncles and father figures especially for boys coming from households where the father is absent. Yet, parents and guardians may view their daughters as sexualized targets by these same neighborhood men. Our research findings echoed the influence of this gender intersection on AT behaviors and perceptions. However, one challenge with our findings was the inability to disentangle the narrative of stranger danger and determine whether this perception was funneled from parents or coming directly from the girls. Regardless of the origination, girls of color may be experiencing limited independent mobility due to these perceptions. Moreover, the ideology of what it means to be a girl may also forestall opportunities for AT. The concept that girls should be “lady-like” and poised whereas boys are expected to be “tough” and rambunctious perpetuates a traditional and faulty rationale that girls should stay indoors or at home. Parallel to this conservative view of girls (e.g., go sit down and read) and boys (e.g., go out and play), is the intention of protecting femininity and allowing for the full expression of masculinity. Several theorists from evolutionary, functional and sociological perspectives have provided extensive conjecture on the roles of men and women. Nevertheless, “gender role theory is grounded in the supposition that individuals socially identified as males and females tend to occupy different ascribed roles within social structures and tend to be judged against divergent expectations for how they ought to behave” [66]. Specifically, girls need to be protected from the outside world and boys are expected to explore the same outside environment. When familial practices affirm the ideology of protecting femininity, the independent mobility, AT and overall physical activity of girls are negatively impacted. This begs the question of what it means to let girls be free.

### 4.3. We’re Moving on Up—Perceptions of Upward Mobility

As mentioned previously, youth from low-income or historically disadvantaged minority families, particularly African American and Latinx youth, experience more transportation obstacles and are more likely to cycle or walk to school compared to Whites or higher income youth [3,4,5,6,7]. Parents of YOC generally have more limited resources to own and maintain a personal vehicle, which results in the higher use of alternative modes of transport including AT among YOC [6,10]. Similarly, participants in this study indicated that AT was a necessity “because my Mom didn’t have a car”. Thus, youth AT behaviors were highly influenced by family fiscal situations that ultimately laid the underpinnings of their AT perceptions. Historically, and to some degree contemporaneously, vehicle ownership in families of color has been identified as a powerful status symbol of upward mobility [67]. As an emblem of success, African American and Latinx groups spend up to 30% more than their White counterparts on visible symbols of success, such as cars, clothing and jewelry [68]. With this strongly rooted attachment to cars, the thought of intentionally engaging in AT may be castigated.

In addition to the availability of individual or family resources as an influential component to AT behaviors and perceptions, the intersectionality of socioeconomic class and AT observed in this study reemphasized previously established associations between neighborhood socioeconomics, perception of safety, and AT. Perceptions of unsafe neighborhoods represent an impediment to physical activity and AT, especially among women, girls, and those living in low-income communities [57,58,59,60]. When girls hail from low-income communities, a proxy for socioeconomic status, which is then intersected with gender and race, it can be theorized that all three of these modalities play a significant role in AT behaviors and perceptions. Findings from our research more than support this inference.

### 4.4. Impacts on Physical Activity and Obesity

African American, Latinx and low-socioeconomic status youth have been found to be more disproportionally affected by the childhood obesity epidemic. However, the environmental and social barriers of AT found in this study, preclude a vital opportunity for physical activity and body weight reduction or maintenance among YOC. While the Metro Silver Line-1 presents a new opportunity for low-income and YOC in Fairfax County to engage in AT, there is a complex issue contending between the want and need for AT opportunities. These youth often need AT because their transport options are limited by resources. Yet, realities and perceptions of safety, as well as, ideologies of social status seem to thwart any real desire for AT among this population of youth who could benefit from the physical activity and obesity prevention advantages of AT.

### 4.5. Strengths and Limitations

There is a paucity of research on public transportation and youth AT in the United States. Moreover, studies from other countries overlooked the exploration of demographic, social, and environmental intersectional influences on youth AT [69,70,71,72]. To the best of our knowledge, this is the first study to apply an intersectionality framework as a tool for examining youth AT. Within a very traditionally quantitative AT research field, this novel and qualitative approach contributed to understanding the structural social and environmental inequalities affecting AT among low-income youth and YOC.

As with most research studies, there are limitations. One limitation of this research involved the inability to recruit participants from outside of Reston and the Reston Teen Center. Challenges with obtaining approval for the recruitment of minors in the local schools and other community centers prohibited opportunities for casting a wider net. Additionally, the AT experiences of the youth reported herein may not be representative of the lived experiences of other 12–15 year old youth in other geographical settings. Fully incorporating intersectionality theory into public health and active living research presents some theoretical, conceptual and measurement challenges. These challenges, however present favorable circumstances for the improvement and enrichment of research quality and the potential to identify and fully uncover social determinants related to health inequities. 

## 5. Conclusions

AT offers an opportunity for youth to increase their daily physical activity and reduce their overweight/obesity risk. This is especially the case for African American, Latinx and low-socioeconomic status youth, who have been more disproportionally impacted by the childhood obesity epidemic in the United States. Although trends also have found these same populations of youth engaging in a higher level of compelled AT due to transportation inequity issues in this country, their experiences of AT have not always been favorable. While the AT benefits are well established in the literature, the environmental and social variables that converge on the AT experiences of low-income youth and YOC have yet to be fully recognized and understood. This study used an intersectionality framework to examine demographic, environmental, and social constructs on AT among a population of Washington D.C. area youth. We found that even though boys expressed more positive perceptions of AT, race, gender, and class operated as AT inhibitors for both boys and girls, with more pronounced negative influences on girls. These qualitative findings offer ways to recognize inequalities so that strategies, such as gamification, can be developed for future interventions that promote a positive AT experience for low-income youth and YOC [73].

## Figures and Tables

**Figure 1 ijerph-16-01313-f001:**
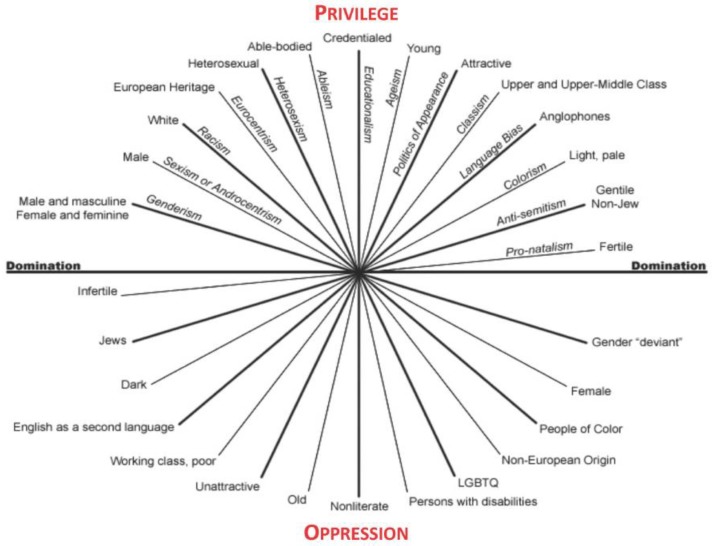
Intersecting Axes of Privilege, Domination and Oppression.

**Figure 2 ijerph-16-01313-f002:**
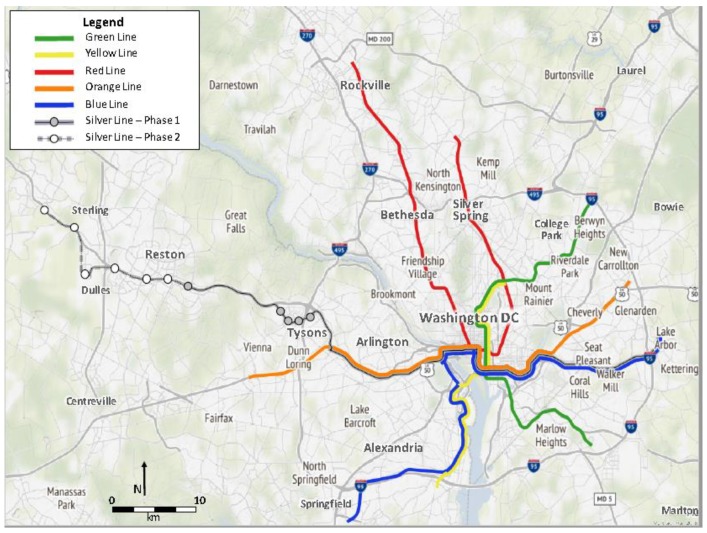
Washington, D.C. Metropolitan Area Transit Authority System.

**Figure 3 ijerph-16-01313-f003:**
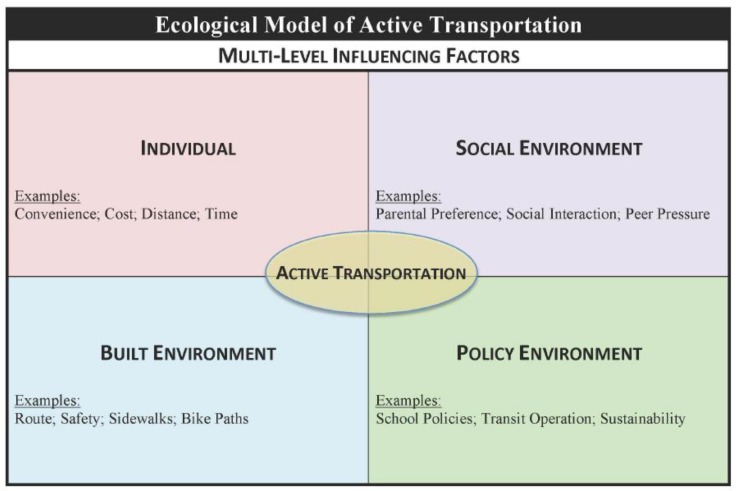
Ecological Model of Active Transportation.

**Figure 4 ijerph-16-01313-f004:**
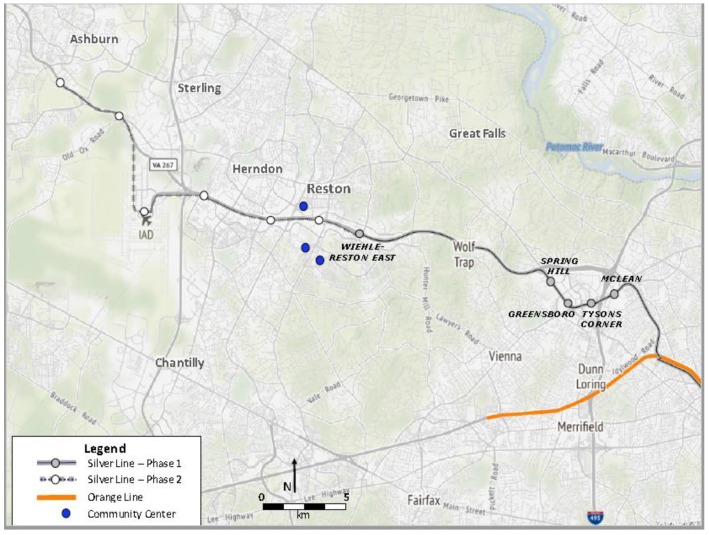
Washington, D.C. Metropolitan Area Transit Authority System-Silver Line-1.

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
