# Peer review of "Between Privilege and Oppression: An Intersectional Analysis of Active Transportation Experiences Among Washington D.C. Area Youth"

_ijerph, 2019, doi:10.3390/ijerph16081313_

Round 1

Reviewer 1 Report

This is an interesting and important study, and it is well-designed. The application of an intersectionality framework has important implications for the examination of demographic, social, and environmental constructs on youth AT, and the conclusions of this study are very clear in this respect, which creates scope for potential improvements. The paper uses a strong theoretical approach the justifications of which are convincing.

As a paper, it is also very well-written.

What follows are a number of minor suggestions for improvement:

-Line 23: "...completed with two groups..."

-Line 24: "...ranging between the ages..."

-Line 60: "...status, has nevertheless been..."

-Line 86: "...barriers, will make youth less inclined..."

-Line 90: "purposeful AT engagement" - Can you explain straight away here why it matters that it is 'purposeful'? This is not immediately obvious, even if you discuss it later in the paper.

-Lines 162-165: Inequity of transportation should lead to more activity for the groups discussed (or at least that seems logical), as they may often not have choice but to use public transport. The paper goes on to complicate this notion considerably, but it may be a good idea to foreshadow that complexity here, so that it's clear where the paper is going from the beginning.

-Lines 246-247 (and also Line 261): Is there an element of 'this is what the researcher wants to hear'? Can you provide some discussion on this potential factor?  

-Line 298: "...or lack of certain..."

-Line 303: "...while speaking to his..."

-Lines 410-414: I think this should be signposted earlier in the paper (see my earlier comment)

Author Response

Reviewer #1,

Please see the attached file which addresses all of your comments and suggested edits.  Thank you for your review.

Best,

Jennifer Roberts

Reviewer 2 Report

This is a very interesting study that reveals significant factors regarding experiences of low-income youth and youth of color with sufficient arguments and discussion. Minor comments follow below:

Reference [1] is nowhere used in text.

Authors are advised to include arguments for modern ways of promoting active transportation, such as Gamification.

Vlahogianni, E.I., Barmpounakis, E.N., 2017. Gamification and sustainable mobility: Challenges and opportunities in a changing transportation landscape, in: Hussein, D. (Ed.), Low Carbon Mobility for Future Cities: Principles and Applications. Institution of Engineering and Technology. http://dx.doi.org/10.1049/PBTR006E_ch12

Kazhamiakin, R., Marconi, A., Perillo, M., Pistore, M., Valetto, G., Piras, L., ... & Perri, N. (2015, October). Using gamification to incentivize sustainable urban mobility. In 2015 IEEE First International Smart Cities Conference (ISC2) (pp. 1-6). IEEE.

Millonig, A., Wunsch, M., Stibe, A., Seer, S., Dai, C., Schechtner, K., & Chin, R. C. (2016). Gamification and social dynamics behind corporate cycling campaigns. Transportation research procedia19, 33-39.

Author Response

Reviewer #2,

Please see the attached file which addresses all of your comments and suggested edits.  Thank you for your review.

Best,

Jennifer Roberts

Reviewer 3 Report

The paper is well written in terms of language which is obvious due to the fact that the authors seem native to the language.

I do not see much research value to the results of study 1) sample size is not significant 2) The research scope is too narrow providing only information on the difference that can be observed by intersectionality framework and authors do not provide enough evidence on how it can be used/utilized in developing interventions. This can be added as a research value to this study.

The focus of the research is not straight forward. A reader gets two different ideas: they are analyzing the impact on AT behavior of specific physical environment change (The Dulles Corridor Metrorail Project), the impact of the intersectional framework on AT behavior. The link between them is not clear. The lines 215 and 216 "Finally, an intersectionality framework examining the demographic, environmental and social intersectional influences on youth AT was incorporated within the analysis." say something about this link but how?

Methodology section can be made clear by mentioning the focussed points of the ecological model of AT.

Discussion section gives a nice overview of the literature and reality but it also postulates the fact that being the inhibitor of AT these Race-class-gender are non-modifiable factors? How do the authors see this? 

Concepts are not/less /defined/explained such as Intersectionality framework is not easy to understand, YOC, Ecological model of AT, JDR. 

Many lines in the manuscripts contain the two spaces after the period e.g. 40, 44, 84, 90, 104, ...... Authors must check the spacing carefully. 

The novelty of the study is questionable as authors themselves refer to few studies: line numbers 84 and 93 for example 

Author Response

Reviewer #3,

Please see the attached file which addresses all of your comments and suggested edits.  Thank you for your review.

Best,

Jennifer Roberts

Round 2

Reviewer 3 Report

Dear authors,

Thank you for improving and making things further clear. 

I agree with the changes.